# Cryo-EM structure of the KvAP channel reveals a non-domain-swapped voltage sensor topology

Xiao Tao, Roderick MacKinnon*

Laboratory of Molecular Neurobiology and Biophysics, The Rockefeller University, Howard Hughes Medical Institute, New York, United States

**Abstract** Conductance in voltage-gated ion channels is regulated by membrane voltage through structural domains known as voltage sensors. A single structural class of voltage sensor domain exists, but two different modes of voltage sensor attachment to the pore occur in nature: domain-swapped and non-domain-swapped. Since the more thoroughly studied Kv1-7, Nav and Cav channels have domain-swapped voltage sensors, much less is known about non-domain-swapped voltage-gated ion channels. In this paper, using cryo-EM, we show that KvAP from *Aeropyrum pernix* has non-domain-swapped voltage sensors as well as other unusual features. The new structure, together with previous functional data, suggests that KvAP and the Shaker channel, to which KvAP is most often compared, probably undergo rather different voltage-dependent conformational changes when they open.

*For correspondence:
mackinn@mail.rockefeller.edu

**Competing interests:** The authors declare that no competing interests exist.

## Introduction

Voltage sensors are integral membrane proteins that 'sense' transmembrane voltage differences. They exist as regulatory domains on voltage-gated ion channels and voltage-dependent phosphatase enzymes (*Hille, 2001*; *Murata et al., 2005*). All voltage sensors have a shared structure consisting of four helical transmembrane segments (*Noda et al., 1984*; *Papazian et al., 1987*; *Tanabe et al., 1987*; *Tempel et al., 1987*; *Jiang et al., 2003a*). The fourth helix, termed S4, contains positive-charged amino acids that move within the transmembrane electrical field as part of a global conformational change (*Noda et al., 1984*; *Tanabe et al., 1987*; *Tempel et al., 1987*; *Aggarwal and MacKinnon, 1996*; *Seoh et al., 1996*; *Long et al., 2005a*; *Long et al., 2005b*; *Long et al., 2007*; *Swartz, 2008*; *Tao et al., 2010*; *Guo et al., 2016*; *Shen et al., 2017*; *Wisedchaisri et al., 2019*; *Xu et al., 2019*). Sensor conformational changes are imparted onto the channel or enzyme to regulate activity.

Atomic structures of several different eukaryotic voltage-gated K⁺ (Kv) channels demonstrate that nature employs two topologically distinct connections between the voltage sensors and the ion channel pore (*Long et al., 2005a*; *Whicher and MacKinnon, 2016*; *Sun and MacKinnon, 2017*; *Tao et al., 2017*; *Wang and MacKinnon, 2017*). Viewed down an axis perpendicular to the membrane, Kv channels are four-fold symmetric, exhibiting one central pore surrounded by four voltage sensors. Each subunit encodes a voltage sensor from transmembrane helices S1-S4 and one fourth of the pore from S5 and S6. The channel tetramer is held together by a 'barrel stave' assemblage of the pore, with the voltage sensors residing on the periphery (*Long et al., 2005a*). The topological distinction between two classes of Kv channels relates to whether the voltage sensors are 'domain-swapped' or not. If four K⁺ channel subunits are designated A, B, C and D, to be domain-swapped means the voltage sensor of subunit A leans against the pore-forming helices of subunit B, B against C and so on around the tetramer, as shown in *Figure 1A*. In non-domain-swapped Kv channels, the voltage sensor of subunit A leans against the pore-forming helices of subunit A, B against B, etc., as

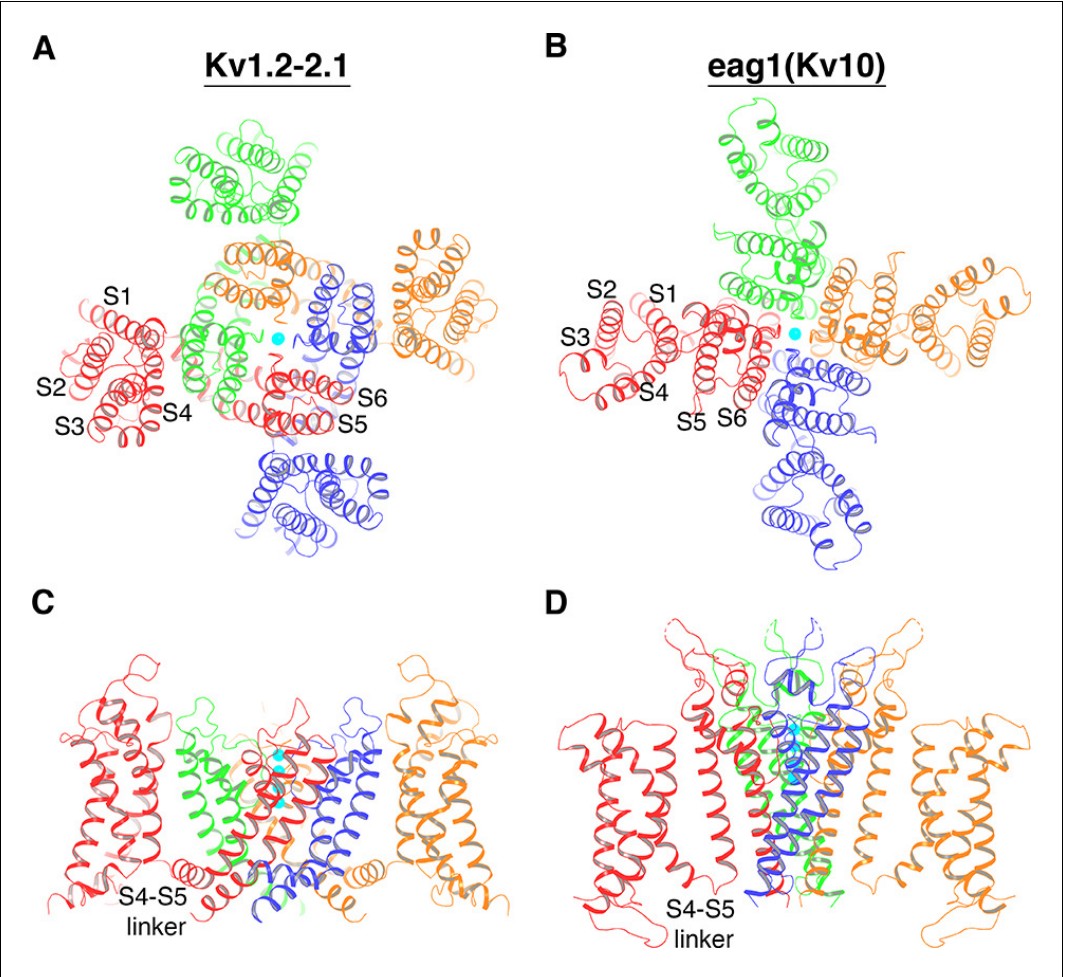

**Figure 1.** Two distinct topological connections between the voltage sensors and the ion channel pore exemplified by Kv1.2–2.1 paddle chimera (PDB: 2R9R) (**A, C**) and eag1 (Kv10) (PDB: 5K7L) (**B, D**) channel, viewed along the 4-fold axis from the extracellular side (**A, B**) or parallel to the membrane with the extracellular side up (**C, D**). The channels are shown as ribbons and each subunit of a tetramer is colored uniquely. In panels C and D, voltage sensors of the front and back subunit are omitted for clarity. $K^+$ ions are shown as cyan spheres.

shown in *Figure 1B*. Kv1-7 channels have domain-swapped voltage sensors; Kv10-12 and Slo1 channels have non-domain-swapped voltage sensors (*Long et al., 2005a*; *Whicher and MacKinnon, 2016*; *Sun and MacKinnon, 2017*; *Tao et al., 2017*; *Wang and MacKinnon, 2017*). Other voltage-gated channels can also be characterized according to this classification: voltage-gated $Na^+$ (Nav) and $Ca^{2+}$ (Cav) channels and other cation channels have domain-swapped voltage sensors (*Payandeh et al., 2011*; *Zhang et al., 2012*; *Guo et al., 2016*; *Wu et al., 2016*; *Shen et al., 2017*; *Yan et al., 2017*; *She et al., 2018*); HCN channels, which are hyperpolarization-activated cation channels, have non-domain-swapped voltage sensors (*Lee and MacKinnon, 2017*).

The difference between domain-swapped and non-domain-swapped topologies is much more than an architectural curiosity. In the former case, the voltage sensor is connected to the pore by an ~20 Å long S4-S5 linker helix, which runs along the channel's cytoplasmic surface and contacts the S6 helix, which forms the gate (*Figure 1C*) (*Long et al., 2005a*; *Long et al., 2007*). Conformational changes within the voltage sensor are transmitted to the gate through the S4-S5 linker helix (*Long et al., 2005b*; *Guo et al., 2016*; *Wisedchaisri et al., 2019*; *Xu et al., 2019*). Channels with non-domain-swapped voltage sensors do not have a helical S4-S5 linker and the mechanism of gate control is unknown (*Figure 1D*).

KvAP is a Kv channel from the thermophilic archaea *Aeropyrum pernix* (*Ruta et al., 2003*). It provided the first atomic structures from a voltage-gated ion channel and the isolated voltage sensor

structure turned out to be the prototype for this domain family (*Jiang et al., 2003a*). Crystallographic analysis of the full-length KvAP channel, however, consistently yielded structures with voltage sensors that were rotated relative to the pore and partially unfolded (*Jiang et al., 2003a*; *Lee et al., 2005*). Together with voltage-dependent accessibility measurements, the KvAP structures led us to propose the 'paddle model', in which S4 moves adjacent to the lipid membrane as part of a helix-turn-helix structure consisting of helices S3b and S4 (*Jiang et al., 2003b*).

Until recently, all voltage-gated channels were thought to be of the domain-swapped variety (*Whicher and MacKinnon, 2016*). For this reason, KvAP was considered representative of the more extensively studied Shaker-like (i.e. Kv1) channels, with domain-swapped voltage sensors (*Long et al., 2005a*). We show here with a cryo-EM structure that this is not the case. In addition to being a non-domain-swapped Kv channel, KvAP has other structural features that distinguish it further from Shaker and most other voltage-gated ion channels. Structural differences between KvAP and Shaker may account for some discrepant results in the study of their voltage sensor conformational changes.

## Results

### Image analysis and map calculation

KvAP was expressed in *E. coli*, extracted in a mixture of lauryl maltose neopentyl glycol (LMNG) and cholesteryl hemisuccinate (CHS), exchanged into digitonin and then purified as a complex with Fab fragments using size exclusion chromatography (*Figure 2—figure supplement 1*). The Fab fragments, which bind to KvAP's voltage sensors, were used to assist the alignment of channels in image processing. Following 2D classification, 734,850 particles were 3D-classified in Relion3 with C1 symmetry (*Figure 2—figure supplement 2*) (*Scheres, 2012*). All classes showed that the voltage sensors were oriented with Fabs projecting towards the extracellular face of the channel. This orientation is compatible with the extracellular accessibility of these Fabs in electrophysiological studies (*Jiang et al., 2003b*). The extracellular orientation of Fabs in cryo-EM images contrasts with the nonnative orientation of KvAP voltage sensors in crystal structures of the full-length channel, in which detergents more dispersive than digitonin were used (*Jiang et al., 2003a*; *Lee et al., 2005*).

While the Fabs were on the extracellular surface in all 3D classes, the precise positioning of the Fabs was variable, as shown (*Figure 2—figure supplement 2B*). The positional variability explains why in both 2D (*Figure 2—figure supplement 3A,B*) and 3D classes (*Figure 2—figure supplement 2B*, *Figure 2—figure supplement 3C*), density for the four Fabs is not constant: in some classes only a single Fab is well aligned, in others 2, 3, or least frequently 4 Fabs are aligned. The variation in Fab orientation is consistent with past studies showing the high degree of mobility of the S3b-S4 paddle region of the voltage sensor (*Ruta et al., 2005*; *Butterwick and MacKinnon, 2010*). To increase resolution of the structure we selected one class, representing 13% of the particles, in which all four Fabs were oriented most similarly with respect to each other. We emphasize that these particles were not fundamentally different than the others, but statistically they happened to have more similarly-oriented Fabs, thus permitting better particle averaging. To analyze the pore and voltage sensors in greater detail, we further applied focused 3D classification in Relion3 while masking the KvAP transmembrane region and Fabs and also applying C4 symmetry (*Figure 2—figure supplement 2B*). The final density map, calculated to 6 Å resolution, showed continuous density for the transmembrane region, density for the Fabs on the extracellular side and a poorly resolved cytoplasmic domain (CTD) on the intracellular side (*Figure 2*, *Figure 2—figure supplement 2*, and *Figure 2—figure supplement 4*). The existence of the cytoplasmic domain, formed by an extension of the C-terminus, was predicted from protein synthesis studies of KvAP (*Figure 2—figure supplement 4C*) (*McDonald et al., 2019*).

### Structure of KvAP

A model corresponding to the crystal structure of the isolated voltage sensor, unmodified, was placed into the density (*Figure 2A,B*) (*Jiang et al., 2003a*). We note that this model is similar to a second structure that we determined independently using NMR (*Butterwick and MacKinnon, 2010*). A model corresponding to the crystal structure of the pore, unmodified, was also placed into the density (*Figure 2C,D*) (*Jiang et al., 2003a*). Small rigid body adjustments of helical segments

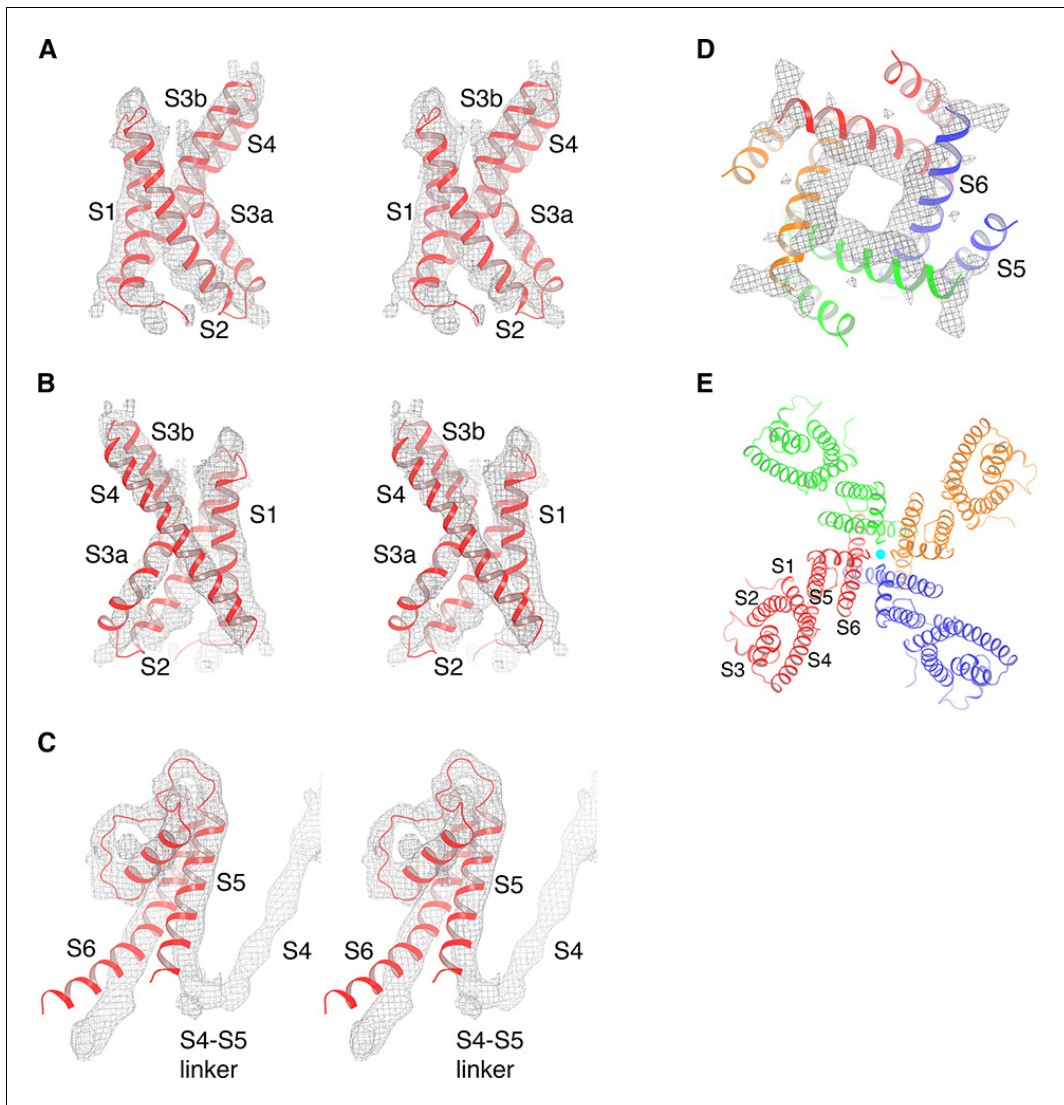

**Figure 2.** Cryo-EM map of KvAP and model generation. (**A**) The isolated KvAP voltage sensor crystal structure (PDB: 1ORS, shown as red ribbons) docked into the cryo-EM map (gray mesh), viewed in stereo, parallel to the membrane with the extracellular side up. (**B**) Same representation as panel (**A**), rotated 180° around the vertical axis. (**C**) The pore domain of a single subunit from the crystal structure of the full-length KvAP (PDB: 1ORQ, shown as red ribbons) docked into the cryo-EM map (gray mesh), viewed in stereo, parallel to the membrane with the extracellular side up. (**D**) The pore domain of the full-length KvAP crystal structure (PDB: 1ORQ, shown as ribbons with each subunit colored uniquely) docked into the cryo-EM map (gray mesh), viewed from the cytoplasmic side. (**E**) Overall structure of KvAP, viewed along the 4-fold axis, from the extracellular side. The channel is shown as ribbons and each subunit of the tetramer is colored uniquely. $K^+$ ions are shown as cyan spheres.

The online version of this article includes the following figure supplement(s) for figure 2:

**Figure supplement 1.** Purification of KvAP-Fab complex.
**Figure supplement 2.** Structure determination of KvAP using cryo-EM.
**Figure supplement 3.** Positional variability of the Fabs.
**Figure supplement 4.** Cryo-EM density map of KvAP reveals the presence of a cytoplasmic domain (CTD).
**Figure supplement 5.** Cryo-EM map from Relion at earlier processing stages.

were made, followed by a round of real-space refinement. The only new model rebuilding at the level of individual amino acids involved four amino acids inside the continuous density connecting S4 to S5. Without the crystal structures it would have been impossible to build more than a poly-alanine model at 6 Å resolution. However, the crystal structures, which were determined at 1.9 Å for the

voltage sensor and 3.2 Å for the pore, conform so well to the experimental map (*Figure 2*) that we know the approximate location of the amino acids in the final model.

An important first conclusion is that KvAP is a non-domain-swapped voltage-gated ion channel, meaning each voltage sensor is contiguous with the pore element from the same polypeptide chain (*Figure 2E*). That this must be the case is evident in the short turn connecting S4 to S5 instead of a long α-helical S4-S5 linker (*Figure 2C*). Therefore, KvAP is topologically like Kv10-12 channels and unlike Kv1-7 channels (*Figures 1* and *2E*).

*Figure 3* highlights a notable structural difference between KvAP and eag1 (Kv10) even though they both share a non-domain-swapped topology. Viewed from the extracellular side, their voltage sensors orient differently with respect to the pore, as shown (*Figure 3A*). The unique orientations result from the manner in which the S1 from the voltage sensor and S5 from the pore interact with each other. In eag1, S1 contacts S5 near the extracellular surface, but is displaced about 15 Å away from S5 near the intracellular surface (*Figure 3C*). In contrast, S1 in KvAP runs parallel to and in contact with S5 throughout its length inside the membrane (*Figure 3B*). Perozo and co-workers analyzed amino acid side-chain accessibility and mobility in membrane-embedded KvAP channels using EPR

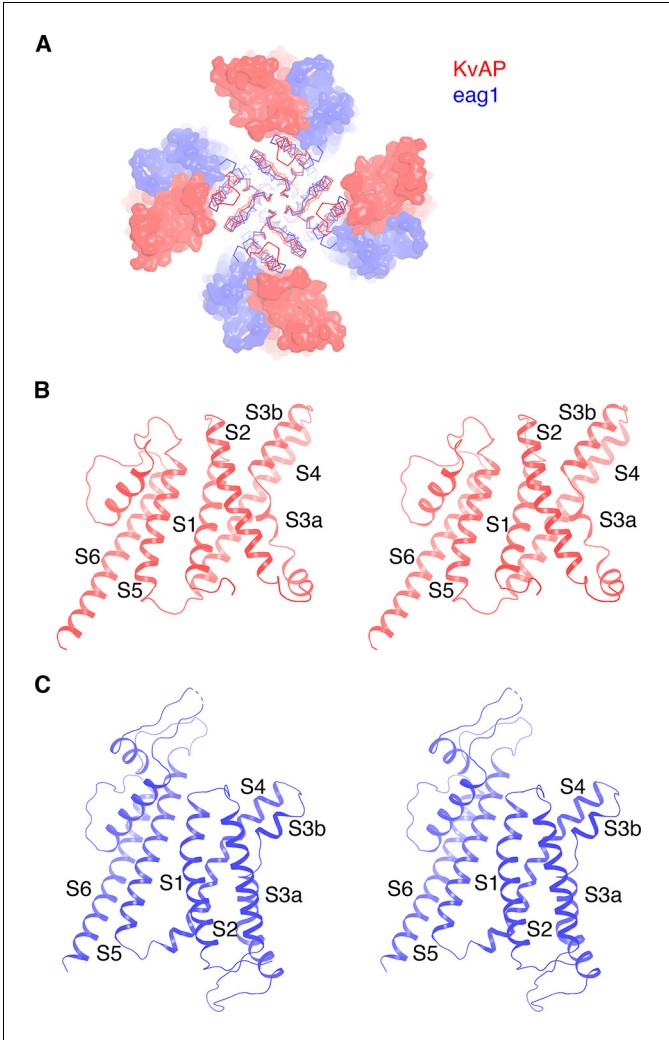

**Figure 3.** Comparison of KvAP and eag1 structures. (**A**) Overlay of KvAP (colored red) and eag1 (PDB: 5K7L, colored blue), viewed along the 4-fold axis, from the extracellular side. The two channels are aligned with respect to the pore domain (shown as Cα traces) and the VSDs are shown as surfaces. For eag1, only the transmembrane region is shown for clarity. (**B**) Ribbon representation of a single subunit of KvAP, viewed in stereo, parallel to the membrane with the extracellular side up. (**C**) Ribbon representation of a single subunit of eag1, viewed in stereo, parallel to the membrane with the extracellular side up. Only the transmembrane region is shown for clarity.

(*Cuello et al., 2004*). Their data, which we have mapped onto the structure of KvAP, support the S1-S5 contact observed (*Figure 4*). Specifically, a continuous surface of low side-chain mobility and low $O_2$ collision frequency, which are both signatures of a buried protein surface, is observed where S1 contacts S5 in the cryo-EM structure. Altogether, the cryo-EM structure is consistent with the EPR data. This consistency suggests that the cryo-EM structure is representative of a conformation of KvAP inside lipid bilayers.

## Conformational state

*Figure 5* shows side views of the KvAP model, excluding the intracellular domain. In electrophysiological studies on KvAP, the membrane must be depolarized first before the extracellularly-applied Fabs can bind (*Jiang et al., 2003b*). In the structure, the helical turn of S3b into S4, the Fab epitope, is exposed to the extracellular surface, compatible with Fab accessibility and a depolarized conformation of KvAP. The voltage sensor is similar in conformation to depolarized voltage sensors in other voltage-gated ion channels (*Figure 5—figure supplement 1*). However, because of the unique interaction between S1 and S5, the S3b-S4 paddle in KvAP resides on the outer-most perimeter of the channel, oriented tangent to the pore domain.

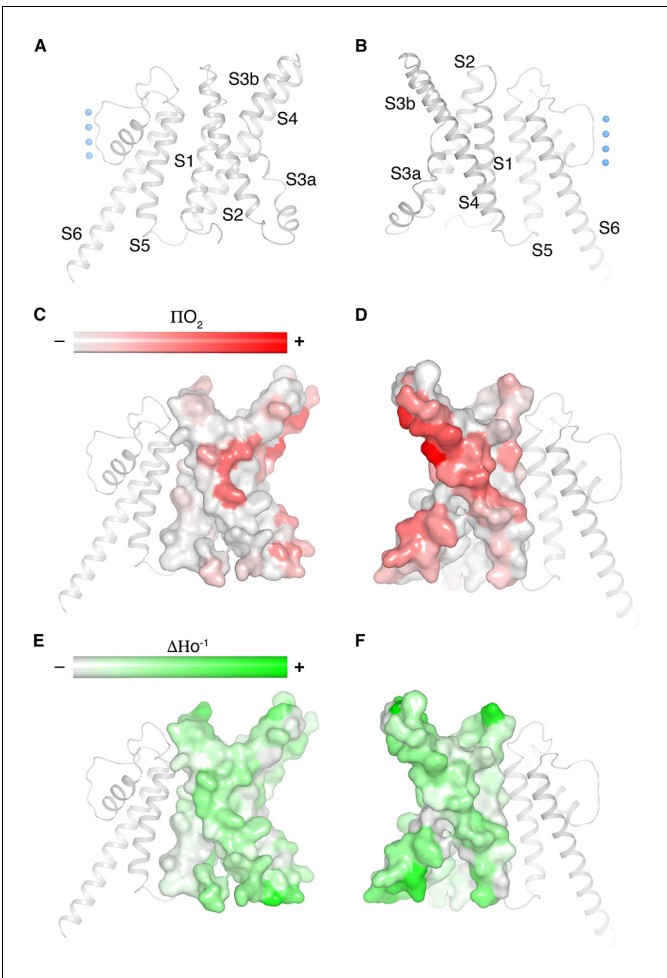

**Figure 4.** Cryo-EM structure of KvAP is consistent with the EPR data in lipid vesicles. (**A,B**) Ribbon representation of a single subunit of KvAP, viewed parallel to the membrane with the extracellular side up. Panel B is rotated 180° from panel A around a vertical axis. (**C–F**) Remapping of EPR data digitized from *Cuello et al. (2004)* onto a single subunit of KvAP (pore domain as gray ribbons, voltage sensor domain as gray surface) in the same views as panels A and B (*Cuello et al., 2004*). Panels C and D show the $O_2$ accessibility (red) and panels E and F show the mobility (green) from reconstituted full-length KvAP.

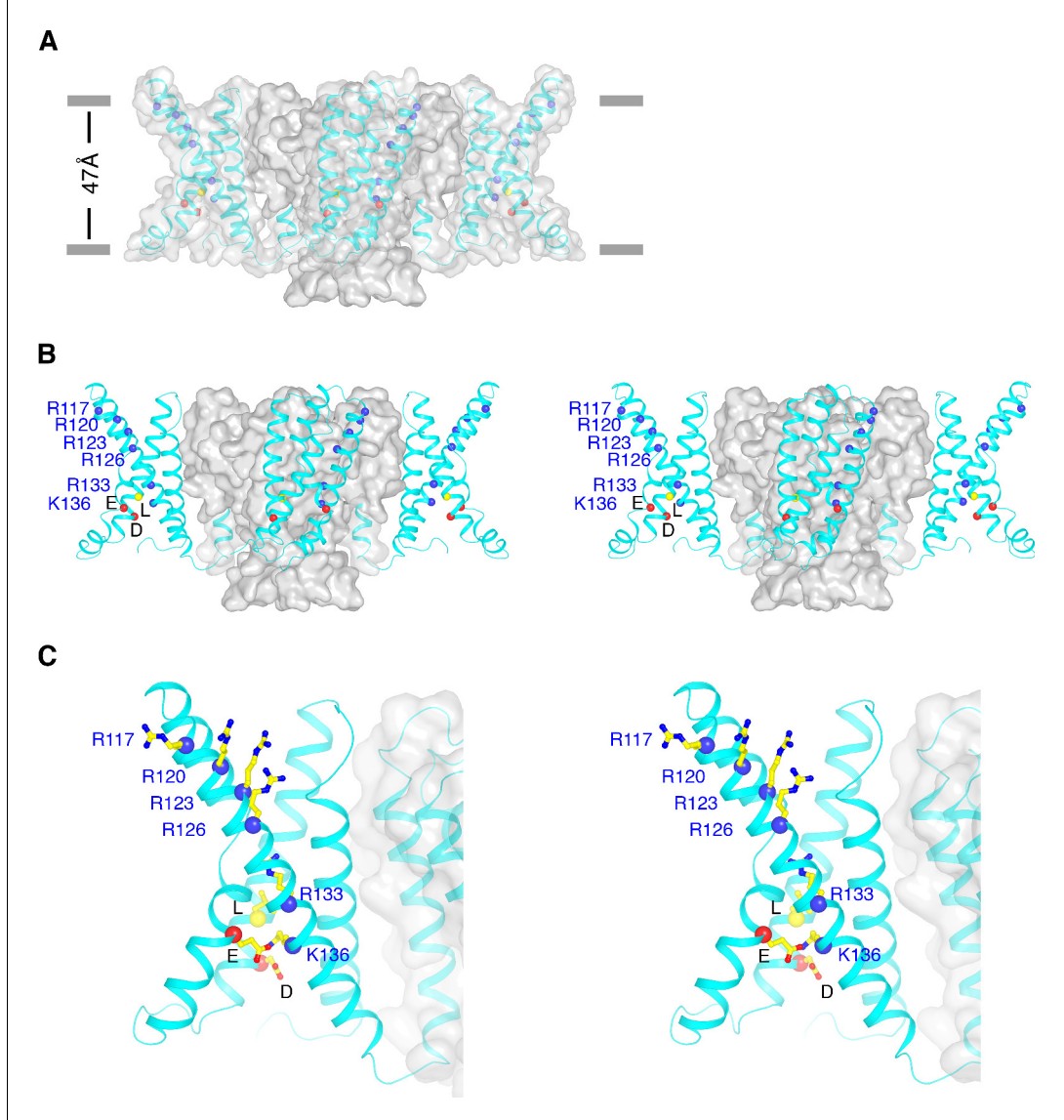

**Figure 5.** Conformational state of KvAP. (**A**) Overall structure of KvAP viewed parallel to the membrane with the extracellular side up. The channel is demarcated by a gray translucent surface and voltage sensors, S4-S5 linkers and intracellular part of S5 are shown as cyan ribbons. Blue spheres show Cα positions of the six positive-charged residues on S4: R117, R120, R123, R126, R133 and K136 from top to bottom. Yellow and two red spheres indicate Cα positions of the gating charge transfer center (yellow: Leu69; Red: Asp72 and Glu93). Gray bars approximate the location of membrane surfaces. (**B**) Stereo view of KvAP in the same orientation as in panel (**A**), with the pore domain shown as a gray surface and the voltage sensor domains, the S4-S5 linkers and the intracellular part of S5 as cyan ribbons. (**C**) Zoomed-in view of one voltage sensor in the same representation as panel (**B**). Side-chains of the six positive-charged residues as well as the gating charge transfer center are shown as sticks and colored according to atom type.

The online version of this article includes the following figure supplement(s) for figure 5:

**Figure supplement 1.** KvAP voltage sensor domain conformation is very similar to other depolarized VSDs.

**Figure supplement 2.** The central conduction pore of the cryo-EM structure of (**A**) KvAP (KvAP_EM), (**B**) the crystal structure of KvAP (KvAP_crystal, PDB: 1ORQ), and (**C**) the Kir2.2 apo crystal structure (Kir2.2_apo, PDB: 3JYC), generated with Hole (*Smart et al., 1996*).

The pore is less open than in the crystal structure of the full-length channel, but more open than in other closed K$^+$ channels (*Figure 2D*, *Figure 5—figure supplement 2*) (*Jiang et al., 2003a*; *Tao et al., 2009*). Functionally, KvAP opens and then inactivates partially following membrane depolarization (*Schmidt et al., 2009*). The conformation of the inner helix gate in this structure might reflect a mixture of open and inactivated channels.

## Remapping of biotin-avidin accessibility data

*Figure 6* maps previous data on the accessibility of tethered biotin to avidin onto the KvAP structure (*Ruta et al., 2005*; *Banerjee and MacKinnon, 2008*). Red and blue spheres show accessibility to the extracellular and intracellular sides of the membrane, respectively. Black spheres show positions inaccessible to either side. With a 1 Å length tether, only superficial sites near the aqueous surfaces are accessible, except at two positions, shown as blue spheres for intracellular accessibility, on the S4 helix. These outlier positions on the structure are more than 15 Å away from the intracellular surface. But the accessibility assay was carried out in membranes under voltage clamp. The result implies that the KvAP voltage sensor can undergo very large voltage-dependent conformational changes to bring these positions close to the intracellular surface. Experiments using longer tethers show results consistent with shorter tethers. Specifically, positions on S4 become exposed to both the extracellular and intracellular sides, depending on the membrane voltage (*Figure 6—figure supplement 1*).

## Discussion

The surprising new finding presented in this paper is that the voltage sensors in KvAP are not domain-swapped. We had assumed incorrectly that KvAP and Kv1 were similar in structure, but they are not (*Lee et al., 2005*; *Ruta et al., 2005*). Through studies of domain-swapped Kv, Nav, Cav and other voltage-gated ion channels we are starting to understand certain aspects of how they work (*Long et al., 2005b*; *Long et al., 2007*; *Tao et al., 2010*; *Guo et al., 2016*; *Shen et al., 2017*; *Wisedchaisri et al., 2019*; *Xu et al., 2019*). We have also seen that even among channels with domain-swapped voltage sensors, the S4 helix apparently moves to different extents to regulate the pore (*Guo et al., 2016*; *Shen et al., 2017*; *Wisedchaisri et al., 2019*; *Xu et al., 2019*). Channels with non-domain-swapped voltage sensors, because they do not have an S4-S5 linker helix, might

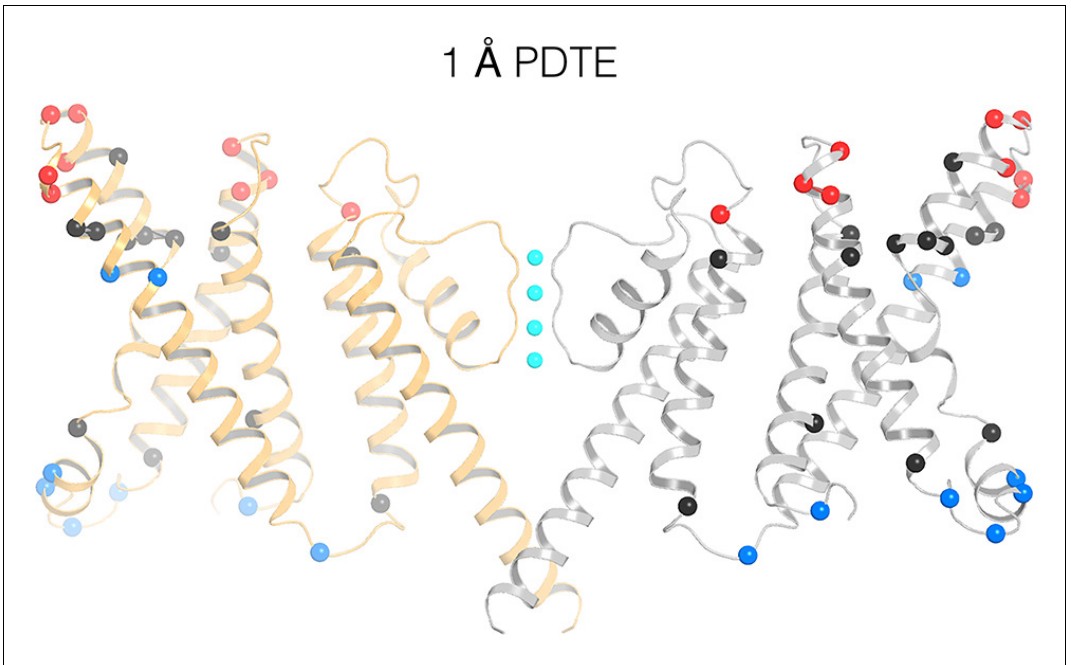

**Figure 6.** Remapping of biotin-avidin accessibility data onto the KvAP cryo-EM structure. Previous data on tethered biotin-avidin accessibility with 1 Å effective tether length is remapped onto the cryo-EM structure of KvAP (*Ruta et al., 2005*; *Banerjee and MacKinnon, 2008*). Only two opposing subunits (gray and orange ribbons) are shown for clarity. Red and blue spheres represent positions accessible to the extracellular and intracellular sides of the membrane, respectively. Black spheres represent positions inaccessible to either side of the membrane.

The online version of this article includes the following figure supplement(s) for figure 6:

**Figure supplement 1.** Remapping biotin-avidin accessibility data onto the KvAP cryo-EM structure.

undergo conformational changes that are not only different in magnitude, but also in character. Obviously, that KvAP has non-domain-swapped voltage sensors is important to the interpretation of functional data, especially those addressing voltage sensor conformational changes (*Cuello et al., 2004*; *Ruta et al., 2005*).

KvAP is an outlier among voltage-gated ion channels, whether domain-swapped or not, in two respects. First, the distribution of positive-charged amino acids on S4 is unique. In other Kv channels the region of S4 that contains gating charge residues – the active region – consists of the strict triplet $(RXX)_n$, where R (arginine) is sometimes replaced by K (lysine) and X stands for a hydrophobic amino acid. More rarely R is replaced by a non-charged polar amino acid, but when this occurs the triplet periodicity almost always persists so that the occurrence of subsequent arginine residues remains in-phase in the sequence. But consider the S4 sequence in KvAP: (RLV)(RLL)(RFL)(RIL)(LII)(SRG)(SKF). Here, we observe a discontinuity between the fourth and fifth arginine residues created by six consecutive uncharged amino acids; this produces a phase shift of the fifth and sixth (lysine) with respect to the first four. The fifth R as well as the following lysine in the sequence are certainly part of the active region of S4, because in the structure the lysine resides inside the gating charge transfer center (*Figure 5C*), which is the site at the center of the voltage sensor that lowers the energy barrier for charged amino acids as they cross the membrane (*Tao et al., 2010*). In other voltage-gated ion channels, S4 adopts a $3_{10}$ helix at the center of the membrane; this secondary structure directs the arginine residues towards the gating charge transfer center. The $\alpha$ to a $3_{10}$ transition may well occur when the region of S4 containing the first four arginine residues approaches the center of the membrane (because they are distributed with triplet periodicity), but it might not occur in the gap between the fourth and fifth arginine residues. Because triplet periodicity is not maintained over the lower half of S4 in KvAP, it is not entirely clear whether properties learned from other Kv channels apply to KvAP.

The second respect in which KvAP is an outlier concerns the gating charge transfer center. KvAP has the typical constellation of negative-charged amino acids to stabilize the positive charge on arginine or lysine. But instead of the phenylalanine found in other Kv channels (357 out of 360 examined), KvAP contains a leucine. The functional significance of this is also unknown.

Our original expectation that KvAP and Shaker-like Kv channels probably undergo similar voltage sensor conformational changes may be incorrect. The present study shows that the voltage sensors in KvAP, in contrast to Shaker, are not domain-swapped. In separate studies in our lab, we have captured the HCN channel, another non-domain-swapped voltage-gated channel, in both depolarized and hyperpolarized conformations (*Lee and MacKinnon, 2017*; *Lee and MacKinnon, 2019*). When depolarized, the voltage sensor in HCN is similar to that in KvAP. But when hyperpolarized, the S4 helix translates toward the cytoplasm and breaks into an L-shape, with the bottom half of S4 forming an interfacial helix. Thus, the voltage sensor in HCN couples to the gate in a fundamentally different manner than in domain-swapped voltage-gated channels studied so far, where the S4-S5 linker forms a permanent interfacial helix (*Figure 1C*). Whether the allosteric communication between the voltage sensor and pore in KvAP will be similar to that observed in HCN remains to be seen.

## Materials and methods

### Key resources table

| Reagent type (species) or resource | Designation | Source or reference | Identifiers | Additional information |
|---|---|---|---|---|
| Gene (*Aeropyrum pernix*) | KvAP | synthetic | | Synthesized at GeneWiz. |
| Recombinant DNA reagent | pET28a | Novagen | 69864 | |
| Strain, strain background (*Escherichia coli*) | BL21(DE3) | Invitrogen | C600003 | |

*Continued on next page*

*Continued*

| Reagent type (species) or resource | Designation | Source or reference | Identifiers | Additional information |
|---|---|---|---|---|
| Chemical compound, drug | 2,2-didecylpropane-1, 3-bis-β-D-maltopyranoside (LMNG) | Anatrace | NG310 | |
| Chemical compound, drug | Cholesteryl hemisuccinate (CHS) | Anatrace | CH210 | |
| Chemical compound, drug | Digitonin | Sigma-Aldrich | D141 | |
| Antibody | anti-KvAP (mouse monoclonal, Fab) | DOI: 10.1038/nature01580 | 6E1 | KvAP teramer: 6E1 Fab = 1: 4.4 molar ratio |
| Commercial assay or kit | Talon metal affinity resin | Clontech | 635504 | |
| Commercial assay or kit | Superdex 200 Increase 10/300 GL | GE Healthcare Life Sciences | 28990944 | |
| Commercial assay or kit | R1.2/1.3 400 mesh Au holey carbon grids | Quantifoil | 1210627 | |
| Software, algorithm | SerialEM | DOI: 10.1016/j.jsb.2005.07.007 | http://bio3d.colorado.edu/SerialEM | |
| Software, algorithm | MotionCor2 | DOI: 10.1038/nmeth.4193 | https://msg.ucsf.edu/software | |
| Software, algorithm | Gctf | DOI: 10.1016/j.jsb.2015.11.003 | https://www.mrc-lmb.cam.ac.uk/kzhang/ | |
| Software, algorithm | Gautomatch | other | https://www.mrc-lmb.cam.ac.uk/kzhang/ | |
| Software, algorithm | RELION-3 | DOI: 10.7554/eLife.18722 | http://www2.mrc-lmb.cam.ac.uk/relion | |
| Software, algorithm | cryoSPARC | DOI: 10.1038/nmeth.4169 | http://www.cryosparc.com | |
| Software, algorithm | FrealignX | DOI: 10.1016/j.jsb.2013.07.005 | http://grigoriefflab.janelia.org/frealign | |
| Software, algorithm | COOT | DOI: 10.1107/S0907444910007493 | https://www2.mrc-lmb.cam.ac.uk/personal/pemsley/coot/ | |
| Software, algorithm | PHENIX | DOI: 10.1107/S2059798318006551 | https://www.phenix-online.org | |
| Software, algorithm | UCSF Chimera | DOI: 10.1002/jcc.20084 | https://www.cgl.ucsf.edu/chimera | |
| Software, algorithm | Pymol | PyMOL Molecular Graphics System, Schrödinger, LLC | http://www.pymol.org | |
| Software, algorithm | HOLE | DOI: 10.1016/s0263-7855(97)00009-x | http://www.holeprogram.org | |

## Cloning, expression, and purification

DNA encoding residues 12–282 of KvAP was subcloned into protein expression vector pET28a (Novagen) between NcoI and BglII restriction endonuclease sites with a thrombin cleavage site between a carboxy-terminal hexahistidine sequence and the channel. Channel protein was expressed in BL21(DE3) cells (Invitrogen). For large scale culture, a single colony was inoculated in LB medium overnight at 37°C and then diluted 1:100 (v:v) into auto-inducing ZYM-5052 medium and further inoculated at 37°C for 24–26 hr (*Studier, 2005*).

The cell pellet from a 2-liter culture was resuspended in 100 ml of lysis buffer (20 mM Tris-HCl pH 8.0, 150 mM KCl, 0.1 mg/ml DNase I, and protease inhibitors including 0.1 µg/ml pepstatin A, 1 µg/ml leupeptin, 1 µg/ml aprotinin, 0.1 mg/ml soy trypsin inhibitor, 1 mM benzamidine, 0.1 mg/ml 4-(2-

Aminoethyl) benzenesulfonyl fluoride hydrochloride (AEBSF) and 2 mM phenylmethysulfonyl fluoride (PMSF)) and then passed through a microfluidizer three times before adding lauryl maltose neopentyl glycol/cholesteryl hemisuccinate (LMNG/CHS) to a final concentration of 10 mM/2 mM. After 2 hr of extraction at room temperature, the lysate was clarified by centrifugation at 39,800 x g for 30 min. Supernatant was then applied to a prepacked 4 ml Talon $Co^{2+}$ affinity column (Clontech) equilibrated in wash buffer (20 mM Tris-HCl pH 8.0, 150 mM KCl, 0.06% Digitonin). Nonspecifically bound protein was washed using 15 mM imidazole added to the above buffer. The channel was eluted with 400 mM imidazole, concentrated using an Amicon Ultra centrifugal filter (MWCO 50 kDa), and then injected onto a Superdex-200 (10/30) column (GE Healthcare) equilibrated with wash buffer. Fractions corresponding to KvAP tetramer (indicated by black ** in *Figure 2—figure supplement 1*) were pooled and concentrated to ~5 mg/ml, mixed with purified 6E1 Fab at a molar ratio of 1:1.1 (KvAP monomer:6E1 Fab) on ice for 0.5 hr, and loaded onto Superdex-200 column again. Fractions corresponding to KvAP-6E1 Fab complex (indicated by red * in *Figure 2—figure supplement 1*) were pooled, concentrated to ~6.5 mg/ml and used for cryo-EM grid preparation.

## Fab preparation

Monoclonal antibody 6E1 (mouse immunoglobulin) against KvAP were raised using standard procedures (*Harlow, 1989*) and purified as previously described (*Jiang et al., 2003a*). In brief, IgGs from mouse hybridoma cell culture supernatant were purified using a Protein A column (Bio-Rad). Fab fragments were generated by papain proteolysis (Worthington) and purified by Q-Sepharose chromatography (GE Healthcare). The hybridoma cell line has not been authenticated or tested for mycoplasma contamination.

## Cryo-EM grid preparation and imaging

3.5 µl of purified protein was pipetted onto glow-discharged Quantifoil Au 400 mesh, R 1.2/1.3 holey carbon grids (Quantifoil). Grids were blotted for 4 s with a blotting force of 1 and humidity of 100% and flash frozen in liquid-nitrogen-cooled liquid ethane using a FEI Vitrobot Mark IV (FEI).

Grids were then transferred to a FEI Titan Krios electron microscope operating at an acceleration voltage of 300 keV. Images were recorded in an automated fashion on a Gatan K2 Summit detector (Gatan) set to super-resolution mode using SerialEM (*Mastronarde, 2005*). Specifically, images were recorded at a super-resolution pixel size of 0.514 Å and defocus range of 0.8 to 2.2 µm, for 10 s with a subframe exposure time of 200 ms in a dose of approximately 7.9 electrons per pixel per second (a total accumulated dose of approximately 75 electrons per $Å^2$ over 50 subframes or approximately 1.5 electrons per $Å^2$ per subframe).

## Image processing and map calculation

Dose-fractionated super-resolution images were 2 × 2 down sampled by Fourier cropping for motion correction with MotionCor2 (5 × 5 patches) (*Zheng et al., 2017*). The parameters of the contrast transfer function were estimated by GCTF (*Zhang, 2016*). Following motion correction,~5 k particles from a subset of the images were interactively selected using Relion to generate templates representing different views for automated particle selection with gautomatch (https://www.mrc-lmb.cam.ac.uk/kzhang/). The autopicked particles were then subjected to 2D classification in Relion to remove false positives as well as particles belonging to low-abundance classes (*Scheres, 2012*). Particles belonging to 2D classes with more constant 4 Fabs were used for initial model generation in cryoSPARC (*Punjani et al., 2017*).

~735 k particle images were selected following 4 rounds of 2D classification in Relion (*Scheres, 2012*). These particles were separated into eight classes using Relion's 3D classification algorithm without a mask. All eight classes showed presence of 4 Fabs on the opposite side of the CTD density, although the positions of Fabs are variable. One of the eight classes, accounting ~13% (~97 k particles) of the total particles, showed most continuous densities for the KvAP channel moiety after 3D auto-refine in Relion imposing C4 symmetry. The resulting map has a resolution of ~8 Å before postprocessing, showing clearly non-domain-swapped S4-S5 linker density (*Figure 2—figure supplement 5*). The refined particle images of this class were subjected to further focused 3D classification masking the KvAP transmembrane region and Fabs while applying C4 symmetry and skipping image alignment. Rotational and translational parameters of particles from two major classes

(~67 k particles) of the focused 3D classification were used as the input for further focused refinement masking around the KvAP transmembrane region and Fabs in FrealignX, during which the resolution of the reference map used for alignment was limited to 8 Å (*Lyumkis et al., 2013*). The final map achieved a resolution of ~5.9 Å as assessed by Fourier shell correlation using the 0.143 cut-off criterion (*Figure 2—figure supplement 2*) and was sharpened using an isotropic b-factor of $-500$ Å$^2$. Symmetry expansion of the above 67 k particles in Relion followed by local refinement in cryo-SPARC slightly improved the map quality (*Scheres, 2012*; *Punjani et al., 2017*).

### Model building and refinement

For model generation, the crystal structure of the isolated voltage sensor (PDB: 1QRS) and a model of the pore from the crystal structure of the full-length KvAP channel (PDB: 1ORQ) were placed into the density. Small rigid body adjustments of helical segments were made, followed by a round of real-space refinement in Phenix (*Afonine et al., 2018*). The only new model rebuilding at the level of individual amino acids involved four amino acids inside the continuous density connecting S4 to S5. Figures were prepared using PyMOL (Molecular Graphics System, Version 2.2.0 Schrodinger, LLC) and Chimera (*Pettersen et al., 2004*).

## Acknowledgements

We thank Mark Ebrahim and Johanna Sotiris at the Evelyn Gruss Lipper Cryo-EM Resource Center at Rockefeller University for assistance in data collection; thank Dr. Joel Butterwick (Rockefeller University) for advice on protein expression, Dr. Xiaochen Bai (UT Southwestern) for advice on data processing; and members of the MacKinnon lab and Chen lab (Rockefeller University) for assistance. This work was supported in part by GM43949. R.M. is an investigator in the Howard Hughes Medical Institute.

## Additional information

### Funding

| Funder | Grant reference number | Author |
|---|---|---|
| Howard Hughes Medical Institute | | Xiao Tao<br>Roderick MacKinnon |
| National Institutes of Health | GM43949 | Xiao Tao<br>Roderick MacKinnon |

The funders had no role in study design, data collection and interpretation, or the decision to submit the work for publication.

### Author contributions

Xiao Tao, Data curation, Formal analysis, Validation, Investigation, Project administration, Writing—review and editing; Roderick MacKinnon, Conceptualization, Resources, Formal analysis, Supervision, Funding acquisition, Validation, Writing—original draft, Writing—review and editing

### Author ORCIDs

Xiao Tao ⓘD https://orcid.org/0000-0002-9381-7903
Roderick MacKinnon ⓘD https://orcid.org/0000-0001-7605-4679

### Decision letter and Author response

Decision letter https://doi.org/10.7554/eLife.52164.sa1
Author response https://doi.org/10.7554/eLife.52164.sa2

## Additional files

### Supplementary files

- Transparent reporting form

## Data availability

The B-factor sharpened 3D cryo-EM density map and atomic coordinates of KvAP have been deposited in the Worldwide Protein Data Bank (wwPDB) under accession number EMD-20924 and 6UWM.

The following datasets were generated:

| Author(s) | Year | Dataset title | Dataset URL | Database and Identifier |
| --- | --- | --- | --- | --- |
| Tao X, MacKinnon R | 2019 | Single particle cryo-EM structure of KvAP | https://www.rcsb.org/structure/6UWM | Protein Data Bank, 6UWM |
| Tao X, MacKinnon R | 2019 | Single particle cryo-EM structure of KvAP | https://www.ebi.ac.uk/pdbe/entry/emdb/EMD-20924 | Electron Microscopy Data Bank, EMD-20924 |

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
