## [Decision Letter]

**Acceptance summary:**

In this study, Tao and MacKinnon determine the cryo-EM structure of KvAP channel in complex with the Fab fragment of a monoclonal antibody. Despite of low resolution around 6 Å, the previously determined high resolution crystal structure of the isolated VSD of KvAP and the pore region of the full-length KvAP structure conform well with the EM map, allowing them to build an accurate model into the density. The structure reveal some unexpected features that distinguish KvAP from most other canonical Kv channels: (1) the channel adopts a non-domain-swapped topology; (2) S4 does not appear to form a 310 helix and has a distinct arrangement of voltage-sensing arginine residues; (3) KvAP contains a Leu instead of a Phe at its gating charge transfer center; (4) the relative position between VSD and pore in KvAP is different from that in both domain-swapped and non-domain-swapped Kv channels; (5) S1 and S5 form extensive contact in KvAP, which is not observed in other Kv channels. These distinct features suggest that KvAP is an outlier among voltage-gated ion channels and its structural mechanism of voltage gating is likely different from that of most canonical voltage-gated K^+^ channels. KvAP was the channel for which the first structures were solved and thus has been widely used as a model protein for studying voltage gating, the findings of this study become particularly important and may provide explanations for some discrepant results in the studies of voltage-dependent conformational changes. Overall, this is a straightforward and informative study that will valuable for understanding the fundamental mechanisms by which voltage-activated ion channels open and close in response to changes in membrane voltage. The editors think you have addressed all of the issues raised by reviewers and we are delighted to be able to publish your work in *eLife*.

**Decision letter after peer review:**

Thank you for submitting your article "Cryo-EM structure of the KvAP channel reveals a non-domain-swapped voltage sensor topology" for consideration by *eLife*. Your article has been reviewed by three peer reviewers, and the evaluation has been overseen by Kenton Swartz serving as both Reviewing Editor and Senior Editor. The following individuals involved in review of your submission have agreed to reveal their identity: Gilman Toombes (Reviewer #1); Youxing Jiang (Reviewer #2); Fred Sigworth (Reviewer #3).

The reviewers have discussed the reviews with one another and the Reviewing Editor has drafted this decision to help you prepare a revised submission.

As you will see from the comments of the reviewers, all three are enthusiastic about publishing your manuscript in *eLife* and reviewer #1 has suggestions for improving the work in revision. Please consider their comments and address their points in preparing a revised manuscript.

Reviewer #1:

In 2003 the MacKinnon lab revolutionized our understanding of voltage-gated ion channels with the first x-ray crystallographic structures of KvAP, a voltage-gated potassium channel from a thermophilic bacteria. However, KvAP appeared to be distorted in the full-length crystal structure and despite extensive structural and functional studies, the native structure of KvAP is unknown. In this work, the authors have used modern single-particle cryo-EM techniques to obtain a seemingly native-state structure of KvAP in which the voltage-sensing domain (VSD) and pore domain of each KvAP sub-unit are in direct contact, unlike canonical voltage-gated potassium channels (Kv1-9) in which each VSD nestles against the pore domain of the neighboring sub-unit.

This is an important result, and the article is exceptionally clear and well written. However, there are a few aspects of the study that could benefit from further discussion.

1) Effect of Resolution on Model:

The authors have clearly accounted for the EM map resolution (~6 Å) during modelling, but it would help a general reader to present whatever cross-checks the authors performed. For example, did the authors manually assess map uncertainties by plotting different sigma level surfaces, or use a more automated approach (e.g. Beckers, M., Jakobi, A.J., and Sachse, C. (2019). Thresholding of cryo-EM density maps by false discovery rate control. IUCrJ 6, 18-33.). Is the direct S4-S5 linker clearly visible in the raw unmasked ~8 Å map of Class I, or only after masking and symmetry expansion steps? What is the worst resolution at which the EM map could exclude a domain-swapped linker "smeared out" by C4 averaging of multiple conformations, especially if the inner gate were to be in a mixture of open and inactivated states (subsection “Conformational State”)? Does the direct linker density still jump out even if one incorrectly uses a domain-swapped KvAP model (e.g. Figure 6, Lee et al., 2005) for masking and refinement?

2) Effects of Detergent and Antibodies on Conformation:

The MacKinnon lab has previously shown that the conformation of voltage sensing proteins, and KvAP in particular, can be very sensitive to the composition of the lipid membrane. In addition, although Jiang et al. (2003) elegantly showed that binding of the 6E1 antibody drives the channel into a non-conducting state this may not be the physiological inactivated state. In this KvAP structure the VSD domain seems to be displaced towards the extra-cellular side of the membrane (relative to the pore domain) whether one looks at the position of aromatic residues in the TM helices or PEG-Biotin accessibility. It may seem crazy to think lipid composition or antibody-protein interactions could switch a 6-TM channel between direct and domain-swapped configurations, but the Sobolevsky lab observed exactly such a transition for TRPV6 following a single point mutation in the S4-S5 linker (Singh, A.K., Saotome, K., and Sobolevsky, A.I. (2017). Swapping Of Transmembrane Domains In The Epithelial Calcium Channel TRPV6. BioRxiv 141523.) Thus, it would be helpful for the authors to discuss whether the conditions used to obtain the structure could influence the domain organization.

3) Consistency of Structure with Published Cysteine Cross-Linking Data:

The MacKinnon lab has previously reported that KvAP can tetramerize through introduced cysteine residues into the VSD and pore (S4 117C / S5 169C; Figure 5D, Lee et al., 2005; S1 T47C / PD V183C; Lee et al., 2009, PLoS Biol 7, e1000047.) These residue pairs were quite close in KvAP domain-swapped homology models, but are quite distant in this direct KvAP structure. A membrane-based assay to excluded a domain swapped architecture would be especially valuable to check that these cross-links are indeed non-specific.

Reviewer #2:

In this study, Tao and MacKinnon determine the cryo-EM structure of KvAP channel in complex with the Fab fragment of a monoclonal antibody. Despite of low resolution around 6 Å, the previously determined high resolution crystal structure of the isolated VSD of KvAP and the pore region of the full-length KvAP structure conform well with the EM map, allowing them to build a pretty accurately model into the density. The structure reveal some unexpected features that distinguish KvAP from most other canonical Kv channels: (1) the channel adopts a non-domain-swapped topology; (2) S4 does not appear to form a 310 helix and has a distinct arrangement of voltage-sensing arginine residues; (3) KvAP contains a Leu instead of a Phe at its gating charge transfer center; (4) the relative position between VSD and pore in KvAP is different from that in both domain-swapped and non-domain-swapped Kv channels; (5) S1 and S5 form extensive contact in KvAP, which is not observed in other Kv channels. These distinct features suggest that KvAP is an outlier among voltage-gated ion channels and its structural mechanism of voltage gating is likely different from that of most canonical voltage-gated K^+^ channels. As KvAP has been widely used as a model protein for studying voltage gating, the findings of this study become particularly important and may provide explanations for some discrepant results in the studies of voltage-dependent conformational changes. In short, this is a simple but informative study that will bring some clarifications to the channel field.

Reviewer #3:

This is an important study that answers questions that surrounded the first X-ray structure of a voltage-gated channel, namely this one. It is particularly satisfying to see the mapping of accessibility and EPR experiments on the new structure.

---

## [Author Response]

Reviewer #1:[…]1) Effect of Resolution on Model:The authors have clearly accounted for the EM map resolution (~6 Å) during modelling, but it would help a general reader to present whatever cross-checks the authors performed. For example, did the authors manually assess map uncertainties by plotting different sigma level surfaces, or use a more automated approach (e.g. Beckers, M., Jakobi, A.J., and Sachse, C. (2019). Thresholding of cryo-EM density maps by false discovery rate control. IUCrJ 6, 18-33.). Is the direct S4-S5 linker clearly visible in the raw unmasked ~8 Å map of Class I, or only after masking and symmetry expansion steps? What is the worst resolution at which the EM map could exclude a domain-swapped linker "smeared out" by C4 averaging of multiple conformations, especially if the inner gate were to be in a mixture of open and inactivated states (subsection “Conformational State”)? Does the direct linker density still jump out even if one incorrectly uses a domain-swapped KvAP model (e.g. Figure 6, Lee et al., 2005) for masking and refinement?

Yes, we studied these maps carefully at different contours and at different stages of map refinement. The non-domain-swapped S4-S5 linker is clearly visible in the raw unmasked ~8 Å map of Class I. A new supplementary figure (Figure 2—figure supplement 5) is now added to demonstrate this fact.

In cryo-EM image analysis, as it was carried out here, information from an atomic model never enters into the map calculation (unlike model-guided phase refinement in X-ray crystallographic map calculation). Thus, neither a correct nor incorrect model is used to modify phases. The maps we show are experimental. We showed the maps for the entire molecule in stereo to answer the questions this reviewer raises. There is no doubt about the connectivity of the non-helical S4-S5 linker and about the absence of a helical, domain-swapping S4-S5 linker. Furthermore, we show in stereo the unmodified models of the pore and voltage sensor from crystal structures to emphasize how well the map conforms to them.

2) Effects of Detergent and Antibodies on Conformation:The MacKinnon lab has previously shown that the conformation of voltage sensing proteins, and KvAP in particular, can be very sensitive to the composition of the lipid membrane. In addition, although Jiang et al. (2003) elegantly showed that binding of the 6E1 antibody drives the channel into a non-conducting state this may not be the physiological inactivated state. In this KvAP structure the VSD domain seems to be displaced towards the extra-cellular side of the membrane (relative to the pore domain) whether one looks at the position of aromatic residues in the TM helices or PEG-Biotin accessibility. It may seem crazy to think lipid composition or antibody-protein interactions could switch a 6-TM channel between direct and domain-swapped configurations, but the Sobolevsky lab observed exactly such a transition for TRPV6 following a single point mutation in the S4-S5 linker (Singh, A.K., Saotome, K., and Sobolevsky, A.I. (2017). Swapping Of Transmembrane Domains In The Epithelial Calcium Channel TRPV6. BioRxiv 141523.) Thus, it would be helpful for the authors to discuss whether the conditions used to obtain the structure could influence the domain organization.

We cannot conclude with certainty what the functional state of this channel is. In fact, no one can ever do that with a structure. Based on comparison to other voltage sensors and to the biotin-avidin accessibility data on KvAP, recapitulated in the paper, we conclude the voltage sensors are depolarized. As we stated, the pore could be activated, inactivated (as we would expect based on electrophysiological studies) or some combination of these states.

We do not think that the voltage sensor is displaced towards the extracellular side abnormally. Please note Figure 5A. Also note that voltage sensor Fabs capture the top of the paddle only upon depolarization, and they associate very rapidly, implying the epitope becomes exposed to solution without hindrance by the membrane (Jiang et al. 2003). We think the structure is in good agreement with these observations. Also, the cryo-EM structure is consistent with independent EPR studies of KvAP in lipid vesicles, shown in the paper.

We think that the interchangeable topologies in mutant TRPV6 channels of a single species is a singular observation and very confusing and the physiological significance unclear. Time will tell what those findings mean. Since such behavior has never been seen anywhere else I do not think our paper is the place to discuss it.

Prior to knowing the existence of Kv channels with non-domain-swapped voltage sensor topology, we modelled KvAP as having a domain-swapped voltage sensor (Lee et al., 2005). This was difficult because it seemed there were not enough amino acids to make S4 and an S4-S5 linker helix. Consequently, S4 had to be very short and the tip of the voltage sensor paddle had to be embedded in the membrane. From the present work we now understand the problem. KvAP has a non-domain-swapped topology. The structure we present here is the first hard evidence for the real topology (i.e. we finally see it).

3) Consistency of Structure with Published Cysteine Cross-Linking Data:The MacKinnon lab has previously reported that KvAP can tetramerize through introduced cysteine residues into the VSD and pore (S4 117C / S5 169C; Figure 5D, Lee et al., 2005; S1 T47C / PD V183C; Lee et al., 2009, PLoS Biol 7, e1000047.) These residue pairs were quite close in KvAP domain-swapped homology models, but are quite distant in this direct KvAP structure. A membrane-based assay to excluded a domain swapped architecture would be especially valuable to check that these cross-links are indeed non-specific.

Both studies (Lee et al., 2005 and Lee et al., 2009) were done in lipid vesicles, not detergents. It is worth pointing out that in our non-domain-swapped cryo-EM model, the Cα-Cα distance between residue 117 (S4) and 169 (S5) on the same subunit is ~35 Å, whereas distance between 117 and 169 on neighboring subunits is ~42 Å. For the other pair, i.e. residue 47(S1) and 183 (pore), the Cα-Cα distance on the same subunit (20.5 Å) and neighboring subunits (21.1 Å). For both cysteine pairs in both models the distances are too far. Neither model is consistent with these disulfide bridges. Unfortunately, when given enough time, as was the case in the Lee et al. studies, disulfide bridges can occur between cysteine residues that are very far apart in protein structures (Careaga and Falke, 1992). Trapping of distant cysteines is a common occurrence unless the kinetics of bridge formation is monitored and shown to be rapid. We now have a much better data set – a structure that explains that KvAP has non-domain-swapped voltage sensors.